

# The natural oscillations in stratospheric ozone observed by the GROMOS microwave radiometer at the NDACC station Bern

Lorena Moreira[1], Klemens Hocke[1], Francisco Navas-Guzmán[1], Ellen Eckert[2], Thomas von Clarmann[2], and Niklaus Kämpfer[1]

[1]Institute of Applied Physics and Oeschger Centre for Climate Change Research, University of Bern, Bern, Switzerland
[2]Karlsruhe Institute of Technology, Institute for Meteorology and Climate Research, Karlsruhe, Germany

*Correspondence to:* L. Moreira (lorena.moreira@iap.unibe.ch)

**Abstract.** A multilinear parametric regression analysis was performed to assess the seasonal and inter-annual variations of stratospheric ozone profiles from the GROMOS (GROund-based Millimeter-wave Ozone Spectrometer) microwave radiometer at Bern, Switzerland (46.95°N, 7.44°E, 577 m). GROMOS takes part in the Network for the Detection of Atmospheric Composition Change (NDACC). The study covers the stratosphere from 50 to 0.5 hPa (from 21 to 53 km) and extends over the period from January 1997 to January 2015. The natural variability was fitted during the regression analysis through the annual and semi-annual oscillations (AO, SAO), the quasi-biennial oscillation (QBO), the El Niño–Southern Oscillation (ENSO) and the solar activity cycle. Seasonal ozone variations mainly appear as an annual cycle in the middle and upper stratosphere and a semi-annual cycle in the upper stratosphere. Regarding the inter-annual variations, they are primarily present in the lower and middle stratosphere. In the lower and middle stratosphere ozone variations are controlled predominantly by transport processes, due to the long lifetime of ozone whereas in the upper stratosphere its lifetime is relatively short and ozone is controlled mainly by photochemistry. The present study shows agreement in the observed naturally induced ozone signatures with other studies. Further, we present an overview of the possible causes of the effects observed in stratospheric ozone due to natural oscillations at a northern mid-latitude station. For instance regarding the SAO, we find that polar winter stratopause warmings contribute to the strength of this oscillation since these temperature enhancements lead to a reduction in upper stratospheric ozone. We have detected a strong peak amplitude of about 5% for the solar cycle in lower stratospheric ozone for our 1.5 cycles of solar activity. Though the 11-year ozone oscillation above Bern is in phase with the solar cycle, we suppose that the strong amplitude is partly due to meteorological disturbances and associated ozone anomalies in the Northern hemisphere. Further, our observational study gave the result that ozone above Bern is anti-correlated to the ENSO phenomenon in the lower stratosphere and correlated in the middle stratosphere.

## 1 Introduction

There is a wealth of possible sources of natural variability in stratospheric ozone. The concentration of stratospheric ozone varies as a result of different factors, some interacting among themselves, through their effects on chemistry and transport. The seasonal variations, given in terms of annual and semi-annual oscillations (AO, SAO), have been studied for many years





(e.g., Ern et al., 2015; Schneider et al., 2005; Cordero and Kawa, 2001; Garcia et al., 1997; Ray et al., 1994; Perliski et al., 1989; Delisi and Dunkerton, 1988; Maeda, 1984). On inter-annual time scales dynamical feedbacks in the Earth System lead to effects of originally tropical phenomena, such as quasi-biennial oscillation (QBO) and El Niño–Southern Oscillation (ENSO), on midlatitude wave structures and wave propagation. Planetary waves play an essential role in driving the zonal mean transport by the Brewer-Dobson circulation (BDC) and eddy mixing processes. This affects the zonal mean meridional transport of trace gases from the tropics to mid- and polar latitudes in the stratosphere and also produces variations in the strength of the polar winter vortices and stratospheric warming events (WMO, 2014). On the other hand, naturally induced ozone variability is also caused by inter-annual changes in solar ultraviolet spectral irradiance (Hood and McCormack, 1992). Many studies regarding the inter-annual variability can be found in the literature (e.g., Manzini, 2009; Fischer et al., 2008; Brönnimann, 2007; Randel and Wu, 2007; Austin et al., 2007; Calisesi and Matthes, 2007; Leblanc and McDermid, 2001; Baldwin et al., 2001; Hollandsworth et al., 1995; Brasseur, 1993). Even though a number of mechanisms have been proposed as interpretations of the natural ozone variations in previously mentioned analysis, there are still open questions in the attribution of the causes to the effects observed in the stratosphere. Moreover, these modes of variability do not always play a roll in isolation, there are numerous examples (e.g., Garfinkel et al., 2015; Gabriel et al., 2011; Gray et al., 2010; Hood et al., 2010; Calvo et al., 2009; White and Liu, 2008; Van Loon and Labitzke, 2000; Garcia et al., 1997) where the interactions are known to occur, throwing more complexity into the understanding of this topic. A better comprehension of the natural oscillations would offer a better recognition and predictability of stratospheric ozone trends. As a matter of fact the knowledge of the natural ozone variations is useful to unmask effects on trends, i.e. to distinguish between natural signals and anthropogenic signals. In addition, the anthropogenic signal can influence the natural signal as for example the recently postulated changes in the Brewer-Dobson circulation (BDC) in response to increasing greenhouse gases (Butchart et al., 2006). Consequently, the understanding of ozone variability is very useful for the detection and attribution of long-term changes.

An analysis of this sort at a single station may offer valuable information, useful not only for the comprehension at regional levels but also for the validation of model simulations. In fact, in Mitchell et al. (2014) the different reanalysis data sets give different results at the extra tropics. Therefore, observational studies can be very helpful to shed light on the subject. In addition, our station can contribute to the understanding of the natural oscillations since there are just a few observational studies (e.g., Nair et al., 2013; Calisesi et al., 2005; Schneider et al., 2005) of naturally induced stratospheric ozone variability in mid-latitudes. The GROMOS ozone radiometer has been performing continuous observations of stratospheric ozone profiles since 1994 above Bern (46.95°N, 7.44°E, 577 m). GROMOS is part in the Network for the Detection of Atmospheric Composition Change (NDACC); hence our more-than-20-year time series are available via http://ftp.cpc.ncep.noaa.gov/ndacc/station/bern/hdf/mwave. Long-term ground-based measurements allow the empirical characterisation of natural cycles in stratospheric ozone as a function of altitude. In Moreira et al. (2015) we applied the multilinear parametric trend model (von Clarmann et al., 2010) to derive the long-term trend in stratospheric ozone above Bern. Here, we optimise this method to determine the basic natural oscillations in stratospheric ozone during the past 18 years. The selection of the time interval was the same as in Moreira et al. (2015), based on the assumption that the concentration of equivalent effective stratospheric chlorine (EESC) peaked in 1997 at midlatitudes (WMO, 2011). The regression model includes a linear term, the annual and semi-annual oscillation,



QBO, ENSO and the solar activity cycle. Many studies use regression analysis to evaluate natural variability in the middle atmosphere (e.g., Mitchell et al., 2014; Hood et al., 2010; Randel et al., 2009; Randel and Wu, 2007; Austin et al., 2007; Hood and McCormack, 1992).

The layout of this study is as follows: the description of the data sources employed is provided in Section 2. Details of the regression technique are given in Section 3. Section 4 presents a short summary of the most important processes for stratospheric ozone production and destruction. Section 5 describes the results of the analysis and provides an overview of the possible causes of the natural oscillations observed in stratospheric ozone. Finally the conclusions are summarised in Section 6.

## 2 Data sources

The present study is based on stratospheric ozone profiles observed by GROMOS. The instrument is a ground-based ozone microwave radiometer which is part of the NDACC. It has been continuously observing the middle atmosphere above Bern, Switzerland (46.95°N, 7.44°E, 577 m above sea level) since November 1994. It measures the thermal microwave emission of a rotational transition of ozone at 142.175 GHz. The altitude range of the retrieved ozone profiles covers 25 to 70 km with a vertical resolution of 8–12 km in the stratosphere. The measurement response between 50 and 0.5 hPa (20 to 52 km) is higher than 0.8 (corresponding to an a priori contribution of less than 20%), therefore the retrieved ozone values at these altitudes are primarily based on the measured line spectrum. For technical details, measurement principle and retrieval procedure on the instrument, see for example Moreira et al. (2015); Peter (1997) and references included therein. The vertical ozone profiles from GROMOS have been validated by means of nearby ozone sondes, ground stations and collocated satellite measurements. Its data set has been used for studies of ozone-climate interaction, middle atmospheric dynamics as well as for long-term monitoring of the stratospheric ozone layer and for the detection of trends (Moreira et al., 2015; WMO, 2014; Studer et al., 2014, 2013; Hocke et al., 2013; Studer et al., 2012; van Gijsel et al., 2010; Keckhut et al., 2010; Flury et al., 2009; Steinbrecht et al., 2009, 2006; Dumitru et al., 2006; Calisesi et al., 2001; Peter et al., 1996; Peter and Kämpfer, 1995).

The data used to analyse the temperature, zonal wind, meridional wind and vertical wind are from the European Centre for Medium–Range Weather Forecast (ECMWF) operational analysis for the given location and time interval. In addition, we have also utilised temperature profiles measured by TEMPERA (TEMPERature RAdiometer) microwave radiometer. This instrument was located at the University of Bern, as well as GROMOS, until the end of 2013, after that it was moved to Payerne, Switzerland (46.82°N, 6.95°E, 491 m above sea level and 40 km southwest of Bern) in the frame of a measurement campaign. TEMPERA is a novel ground-based microwave radiometer that measures the thermal radiation emitted by oxygen in the microwave spectrum region in a frequency range from 51–57 GHz. This radiation contains information on the atmospheric temperature. This is the first radiometer that provides temperature profiles in the troposphere and in the stratosphere at the same time. In this study we only use the stratospheric temperature profiles with an altitude range of 18–50 km (70 to 0.7 hPa) and with a vertical resolution of 15 km, retrieved with a measurement response higher than 0.6. The TEMPERA radiometer is described in more detail in (Navas-Guzmán et al., 2015, 2014; Stähli et al., 2013).





## 3 Regression analysis

The regression analysis of the time series of ozone monthly means from GROMOS and the other monthly mean products (temperature, zonal wind, meridional wind and vertical wind) from ECMWF for the period from January 1997 to January 2015 has been carried out using the following multilinear regression function:

$$\hat{y}(t) = a + b \cdot t + c_1 \cdot qbo_1(t) + d_1 \cdot qbo_2(t) + e \cdot F10.7(t) + f \cdot MEI(t) + \sum_{n=2}^{m} (c_n \cdot \sin(\frac{2\pi \cdot t}{l_n}) + d_n \cdot \cos(\frac{2\pi \cdot t}{l_n})) \tag{1}$$

$t$ is the time, and $a$ and $b$ are the constant term and the linear trend of the fit. The QBO indices are $qbo_1$ and $qbo_2$, these terms were implemented by using the normalised Singapore zonal winds at 30 and 50 hPa. These are provided by the Free University of Berlin via http://www.geo.fu-berlin.de/met/ag/strat/produkte/qbo/index.html. The F10.7 term is the normalised time series of the solar radio flux at 10.7 cm which is a proxy of the solar activity cycle. Moreover, the MEI term represents the normalised Multivariate ENSO index (MEI) time series used to identify the ENSO variability during the regression analysis. Both indices are available from www.esrl.noaa.gov/psd/data/climateindices/list. The sum term comprises 2 sine and cosine functions with the period length $l_n$, which represent the annual and semi-annual cycle. The usage of sine and cosine functions give access to the amplitude and the phase of each harmonic. On the contrary to the harmonics and QBO, the multilinear regression model has no access to the phase of the solar activity cycle and ENSO. This results in the detection of the instantaneous response of ozone to the solar activity cycle and ENSO. Nevertheless, we already know from the literature (e.g., Ineson and Scaife, 2009; Manzini, 2009; Brönnimann et al., 2007; Steinbrecht et al., 2004; Sassi et al., 2004; Brönnimann et al., 2004; Lee and Smith, 2003) that the impact of the solar activity cycle and ENSO in ozone have a certain time lag. In order to avoid this problem the fitting of these terms was made with a time delay (1–year for the solar activity cycle and 1–season for the ENSO). The coefficients $a$, $b$, $c_1$, $c_2$, $c_3$, $d_1$, $d_2$, $d_3$, $e$ and $f$ are fitted to the monthly means using the method of von Clarmann et al. (2010). The uncertainties of the monthly means are also required for the regression analysis. For more details on this we refer to Moreira et al. (2015).

## 4 Stratospheric ozone

In this section, we give a brief overview of the most important processes for the photochemistry, chemistry and transport of stratospheric ozone. In 1930, Chapman postulated that ozone is formed by the photolysis of $O_2$ at wavelengths shorter than 240 nm (Reaction R1), immediately followed by the recombination of atomic oxygen with molecular oxygen and any mediating air molecule M via the three–body Reaction R2 (Crutzen and Oppenheimer, 2008). Ozone is removed locally by both transport and chemical processes (Yang et al., 2006). Chapman proposed the photolysis of ozone (Reaction R3 and Reaction R4) and its recombination with atomic oxygen (Reaction R5) to balance the production of ozone. The photodissociation of ozone leads to formation of oxygen atoms in either their ground state ($^3P$) or in their first excited state ($^1D$) (Brasseur and Solomon, 2005).

$$O_2 + h\nu \rightarrow O + O \tag{R1}$$





$$O_2 + O + M \rightarrow O_3 + M \tag{R2}$$

$$O_3 + h\nu(\lambda \geq 320\text{nm}) \rightarrow O_2 + O(^3P) \tag{R3}$$

$$O_3 + h\nu(\lambda \leq 320\text{nm}) \rightarrow O_2 + O(^1D) \tag{R4}$$

$$O + O_3 \rightarrow O_2 + O_2 \tag{R5}$$

Because atomic oxygen and ozone molecules are rapidly interconverted, it is useful to consider both as a family, the odd
oxygen family. Since a typical time scale for meridional transport is of the order of months, the relevant quantity in this context
is the concentration of odd oxygen, not that of its components. The lifetime of odd oxygen in a parcel of air is much longer
than the lifetime of an individual O atom or $O_3$ molecule. The chemical lifetime of odd oxygen ranges from weeks at 30 km
to a year at 20 km. Therefore, odd oxygen has a sufficient long lifetime to be influenced by meridional transport processes
(Brasseur and Solomon, 2005). Consequently transport processes, as for example the Brewer-Dobson circulation, contribute to
the stratospheric ozone distribution.

The amount of $O_x$ that is made up by $O_3$ relative to the amount of $O_x$ made up of O atoms is known as the partitioning of odd
oxygen. The partitioning depends upon the photolysis rate of ozone, the $O + O_2$ reaction rate coefficient and the air density.
The photolysis rate generally depends on the absorption cross-section of ozone and the number of incident UV photons. The
number of photons in turn depends upon a number of other parameters: altitude, latitude, season and time of day. All of these
parameters implicitly depend on the solar zenith angle. As the seasons change from winter to summer, the solar zenith angle
decreases. Therefore, the path that UV photons must travel is shorter in summer than in winter. Consequently, the photolysis
rate coefficients at the upper stratosphere–lower mesosphere have maximum values during summer, when the path length is
shorter and minimum values during winter when the path length is longer. On the other hand, the rate of photolysis depends on
both the number of UV photons and the number of ozone molecules available to interact with photons.

## 5  Results and discussion

### 5.1  Amplitudes of the natural oscillations

The aim of a regression study is to reproduce the evolution through time of the variable under assessment by means of a linear
combination of basic functions. To achieve this goal the regression model includes basic functions representing, in our case,





the solar activity cycle, the El Niño–Southern Oscillation (ENSO), the quasi-biennial oscillation (QBO) and the annual and semi-annual oscillation. As a consequence of using them to fit the ozone monthly means during the regression analysis, we may also use them to quantify the natural variability of ozone. The left panel of Figure 1 shows the amplitude of the regression coefficients of these terms in ppm, along with the ozone mean profile (green line) divided by 10 in order to plot all these

quantities together, whereas in the right panel the amplitudes are plotted in percent. The magenta line represents the annual oscillation (AO), the semi-annual oscillation (SAO) is the cyan line, the orange line is the QBO, ENSO is the red line and the solar radio flux at 10.7 cm (F10.7) is represented by the blue line. The amplitude of the AO dominates at 10 and 2 hPa whereas the SAO has its maximum at 3 hPa. Near 3 hPa at mid-latitudes, the magnitude of the SAO amplitude is larger than the magnitude of the AO amplitude. This effect is also observed by Perliski et al. (1989). After the annual cycle, the solar

variability seems to be the largest source of variation in ozone, exhibiting its influence around 20 hPa. However, the observed 11–year oscillation in stratospheric ozone could be influenced by interfering processes which we will discuss later.

## 5.2   Annual Oscillation (AO)

In the middle stratosphere our observations indicate maximum ozone concentrations in spring–summer and a minimum during the autumn–winter time. The amplitude of the maximum at 10 hPa (32 km) is of the order of 16% (blue line in Figure 2–right

panel) or around 1 ppm (magenta line in Figure 1–left panel). Otherwise in the upper stratosphere during the spring–summer period we observe a minimum and a maximum in autumn–winter time. This maximum has a peak amplitude around 0.6 ppm (magenta line in Figure 1–left panel) or over 16% (blue line in Figure 2–right panel) over 2 hPa (42 km). Our results are in agreement with others observational and modelling studies (Eckert et al., 2014; Calisesi et al., 2005; Schneider et al., 2005; Perliski et al., 1989). For instance, Schneider et al. (2005) obtained a maximum AO-amplitude of the order of 15% at 26

km and around 18% at 40 km with observations between 1995 and 2002 from the Bordeaux microwave radiometer. Further, we are in agreement with the results of Eckert et al. (2014), who used the same regression technique as us, but data from MIPAS on ENVISAT for the period of July 2002 to April 2012. The seasonal variation of ozone can be understood through the partitioning of odd oxygen. Due to the higher flux of short wave radiation in summer more odd oxygen would be expected in the upper stratosphere. The main reason for the minimum ozone in summer in the upper stratosphere is known to be the

temperature-dependent photochemistry. The temperature-induced ozone change is principally dominated by the $NO_x$, $ClO_x$, and $HO_x$ catalytic ozone destruction cycles (Flury et al., 2009) and also by Reaction $R5$ (Schanz et al., 2014) which is sensitive to temperature variations. As a result, ozone depletion is higher in summer than in winter at the stratopause as we can see in Figure 2; the annual ozone maximum around 2 hPa occurs in winter. This feature shows the anti-correlation of ozone and temperature in the stratopause region. After the equinox, when the temperatures start to be warmer the odd-hydrogen and odd-

oxygen reactions proceed faster and ozone is destroyed more rapidly. During winter time, when the temperatures are lower, these reactions proceed more slowly causing the ozone density to increase (Perliski et al., 1989).

    Below the upper stratosphere the anti-correlation of ozone and temperature breaks down, partly because of the small concentration of atomic oxygen and the high air density at lower altitudes where atomic oxygen immediately converts into ozone by Reaction R2. Indeed, Figure 2 clearly shows a phase reversal of the ozone-AO at 3 hPa, where the partitioning of odd oxygen



is balanced by the change from more abundance of atomic oxygen above 3 hPa to more abundance of ozone below 3 hPa. Perliski et al. (1989) suggested that the dominant cause of annual ozone amplitude is the annual variation in the production rate of odd oxygen. In the middle and lower stratosphere, the production rate of odd oxygen is roughly equal to the production rate of ozone because of Reaction R2. This explains the summerly ozone maximum when the production rate of odd oxygen

is maximal. Figure 1 (magenta line–left panel) and Figure 2 (blue line) show that the annual ozone maximum at 10 hPa occurs during early summer. The red line in Figure 2 represents the phase of the ozone-AO given as the month of maximum, and in the ozone-AO maximum at 10 hPa we can clearly see the phase (red line) around 5–6 months i.e. May–June (early summer). This summerly ozone maximum is also influenced by the meridional transport of ozone-rich air from the tropics (Cordero and Kawa, 2001).

**5.3  Semi-annual Oscillation (SAO)**

The semi-annual oscillation (SAO) has a period of 6 months. The ozone-SAO amplitude maximum is around 3 hPa (40 km), and exhibits its maximum of slightly over 0.4 ppm (blue line in Figure 3–left panel) or around 7% (cyan line in Figure 1–right panel). Calisesi et al. (2005) obtained the same order of magnitude for the amplitude of ozone-SAO calculated through an iterative spectral analysis with data from GROMOS for the period 1994–2004. By using the same regression method and data

(July 2002–April 2012) from MIPAS Eckert et al. (2014) got an ozone-SAO amplitude for our latitude around 0.3 ppmv at 40 km. Huang et al. (2008) showed an amplitude of ozone-SAO of the order of 0.3 ppmv around 40 km and at 40°N latitude with measurements from SABER for the period 2002–2005. On the other hand, Perliski et al. (1989) found an SAO amplitude of about 0.5 ppmv around 3 hPa with 9–years (October 1978–September 1987) of SBUV ozone mixing ratio data. The middle panel of Figure 3 shows the phase of ozone-SAO, the month of the maximum is between 3.5 to 4 months at 3 hPa, within

the period of the SAO. This can be confirmed in the right panel of this Figure 3, where the ozone-SAO amplitude against the months of the year is represented. These peak amplitudes coincide in time with the equinoxes, whereas the minima coincide with the solstices. The most interesting point about the ozone-SAO is that its maximal amplitude is located in the same altitude region (upper stratosphere) as the maximal anti-correlation between ozone and temperature (Calisesi et al., 2001).

In order to understand the ozone-SAO, we have also performed the regression analysis to temperature, zonal wind, merid-

ional wind and vertical wind, all of them from the ECMWF operational analysis data over Bern. Since the purpose of this study is to explain the natural variations of ozone only, we are not going to provide explanations for the ECMWF products used but we take them as given. Figure 4 shows the semi-annual cycle at 3 hPa along the year. The amplitude of the temperature (red line), zonal wind (orange line), meridional wind (cyan line), vertical wind (green line) and ozone (blue line) -SAO are plotted together in order to have an overview of which parameter or parameters play a major role in the ozone-SAO. In Figure

4 the ozone-SAO amplitude is multiplied by 10. The ozone-SAO is one month out of phase with that of temperature in the upper stratosphere. This is consistent with ozone photochemistry (Ray et al., 1994). At 3 hPa, the first positive maximum of ozone occurs at the beginning of March, one month after the time of the first SAO-temperature maximum. The stratopause zonal wind-SAO is characterised by the occurrence of easterly winds during the solstice seasons and westerly winds during the equinoxes. The warm phase descends with the easterly shear zone in the solstice season. The cold phase descends with





the westerly shear zone in the equinox season. The zonal wind reversal of the SAO from westward to eastward wind is likely driven by gravity waves. Otherwise, zonal wind reversal from SAO eastward to westward wind is assumed to be mainly driven by horizontal advection and meridional momentum transport of extratropical planetary waves (Ern et al., 2015). High-speed Kelvin wave and gravity wave propagation into the upper stratosphere and the resultant deposition of eddy zonal momentum is

thought to be the most significant forcing of the westerly acceleration (Ray et al., 1994). The out of phase relationship between temperature and vertical wind (upwelling (w>0), green line in Figure 4) is expected due to adiabatic cooling associated with the ascent of ozone-rich air. Therefore, the ozone maximum is found during the equinoxes at 3 hPa, as shown in Figure 4.

Easterly accelerations, on the other hand, appear to be forced meridionally by planetary waves and meridional advection (Ray et al., 1994). During wintertime planetary wave breaking in the upper stratosphere is observed, producing poleward trans-

port and a downward flow (w<0, green line in Figure 4). The downwelling, through air compression, yields an increase in the temperature. The anti-correlation of ozone and temperature is maximal around 3 hPa (upper stratosphere) during winter over Bern (Calisesi et al., 2001). In Figure 5, we observe that an increase in temperature of around 20–30 K causes a decrease in ozone volume mixing ratios in winter of the order of 1–1.5 ppmv. Figure 5 shows the ozone VMR from GROMOS (blue line), the temperature from TEMPERA radiometer (black line) and the temperature from ECMWF operational data (red line) near 3

hPa for the period between January 2012 and January 2015. The good agreement between the temperature from TEMPERA and from ECMWF at this altitude is clearly seen in this Figure 5. We observe the two-peaked ozone curve each calendar year (ozone-SAO), the minima in ozone during wintertime and summertime and the maxima during spring and autumn. Regarding the temperature we notice the fluctuations during wintertime, the increase in spring and summer time and the decrease after the summer solstice. The aforementioned anti-correlation of ozone and temperature is clearly visible, displayed by the upper

stratospheric warming events and the associated ozone decrease in winter. These stratospheric warming events in winter are related to planetary wave breaking (PWB). Planetary waves can break and cause disruptions to the polar vortex and rapid warmings of the stratosphere. The occurrence of warming events are even more frequent in the upper stratosphere compared to the mid-stratosphere as the climatology study of Greer et al. (2013) showed. Thus, the upper stratospheric warmings contribute to the temperature-SAO and the latter contributes to the ozone-SAO. A possible role of stratospheric warmings for the genera-

tion of the ozone-SAO at mid-latitudes, previously mentioned by Perliski et al. (1989) and Maeda (1984). Calisesi et al. (2001) and Flury et al. (2009), showed that the strongest oscillations of ozone above Bern are because of displacements of the polar vortex by planetary wave breaking. The temperature-induced ozone change is mainly dominated by the $NO_x$ catalytic ozone destruction cycle, Reaction R5 and in lesser extent by Reaction R2, which are sensitive to temperature variations (Schanz et al., 2014; Flury et al., 2009). The rate of $NO_x$ cycle and Reaction R5 increases when the temperature rises whereas the rate of

Reaction R2 slows down as the temperature increases. Furthermore, the observed summertime amplitude dip in ozone (blue line in Figure 4) is also due to warming, in this case the increase of temperature in summer. The ozone destruction rates are accelerated during the spring as the temperature rises, causing the observed amplitude drop in ozone after March.





### 5.4 Quasi-biennial Oscillation (QBO)

The quasi–biennial oscillation (QBO) dominates the variability of the equatorial stratosphere and is easily identified as an alternation of descending westerly and easterly wind regimes, with a variable average period length of approximately 28 months. Even though the QBO is a tropical phenomenon, it affects the stratospheric flow from pole to pole and consequently some chemical constituents such as ozone are affected through modulation of extratropical wave propagation induced by the QBO (Baldwin et al., 2001). The QBO was implemented in the multilinear parametric trend model (von Clarmann et al., 2010) using the Singapore zonal winds at 30 and 50 hPa as a proxy, which are approximately phase-shifted by a quarter period so that they are sine and cosine functions of the same period (28.8 months) (Eckert et al., 2014). Their combination can emulate any QBO phase shift (Kyrölä et al., 2010). Stiller et al. (2012) were the first to use the QBO proxies in this multilinear parametric trend model. In Figure 6 and also in Figure 1 (orange line) we can observe that the amplitude of the ozone-QBO maximum is located around 0.15 ppmv (not exceeding 3%) near 30 hPa (24 km). Our results are relatively in agreement with those of Calisesi et al. (2005), with ozone-QBO amplitudes of about 5% at 25 km for the period 1994–2004. These findings also agree with those by Eckert et al. (2014) where they obtained an ozone-QBO amplitude of slightly over 0.2 ppmv in our latitude region at 25 km between 2002–2012. During the easterly (westerly) shear zone of the Singapore zonal winds the amplitude of ozone-QBO observed in Figure 6 is positive (negative). This effect is related to the QBO–induced meridional circulation and confirms the mid-latitude ozone-QBO out of phase relationship with the equatorial ozone-QBO. The amplitude of the equatorial ozone-QBO is positive (negative) during the westerly (easterly) shear zone (Leblanc and McDermid, 2001; Baldwin et al., 2001; Hollandsworth et al., 1995).

The meridional circulation affects chemical tracers such as ozone and gives rise to strong ozone-QBO signals in such tracers at all latitudes (Baldwin et al., 2001). Some models and observational studies (Leblanc and McDermid, 2001; Baldwin et al., 2001; Hollandsworth et al., 1995) found that the downward (upward) vertical motion in the equatorial westerly (easterly) shear zone induces an increase (decrease) in ozone whereas in the mid-latitudes it induces a decrease (increase) in ozone, which we observe in our study.

### 5.5 Solar activity cycle

The solar radiation between 200 and 240 nm is primarily responsible for the formation of ozone in the stratosphere. Changes in solar UV spectral irradiance directly modify the production rate of ozone in the upper stratosphere. The solar activity cycle is dominated by an 11–year solar cycle. The solar activity cycle effect on upper stratospheric ozone is a direct consequence of heating and photochemistry (Reaction R1 and R2 and $NO_x$ cycle). The lower stratospheric response in ozone occurs mainly by means of a dynamical response to solar UV variations (WMO, 2014). Further, the 11–year solar cycle influence on planetary wave propagation will influence the strength of the polar vortex as well as the strength of the mean meridional Brewer-Dobson (BD) circulation since this is forced by planetary wave transfer of momentum. Therefore, a solar cycle variation in ozone amount is to be expected (Mitchell et al., 2014). In order to take into account the effect of solar activity cycle in the stratospheric ozone, a time series of the normalised solar radio flux at 10.7 cm, which is a proxy of the solar activity, was fitted to





the ozone monthly means during the regression analysis. Stratospheric ozone has a delayed response to the solar activity cycle (e.g., Steinbrecht et al., 2004; Cunnold et al., 2004; Lee and Smith, 2003). Therefore, in order to estimate this time lag we checked the time series of total ozone above Arosa (Switzerland) for the 4–solar activity cycles from 1953 to 1996. Generally, the ozone maxima seem to be lagged around 1–year after the solar maximum regarding the time series of solar radio flux at

10.7 cm. Further, Steinbrecht et al. (2004) mentioned a lag of about 1–year behind the maxima of the 10.7 cm solar radio flux time series for ozone profiles measured at Hohenpeissenberg (Germany). Accordingly, we have performed the regression analysis by shifting the solar activity cycle proxy 1–year.

In Figure 7 is represented the normalised time series of the monthly mean 10.7 cm solar radio flux (green line) along with the
amplitude of the response in ozone to this proxy between January 1997 and January 2015. We can observe the increase in ozone volume mixing ratio in the stratosphere during periods of increased solar activity, and the opposite during the solar minimum. The amplitude of the 11–year response in ozone is of the order of 5% or 0.25 ppmv in the lower stratosphere and around 3% or smaller than 0.1 ppmv in the stratopause (blue line in Figure 1). The result in the lower stratosphere is in agreement with the study of Calisesi et al. (2005), in which they studied the natural variability of stratospheric ozone from GROMOS for the time
interval between 1994 and 2004 with an iterative spectral analysis. Additionally, Steinbrecht et al. (2004) found a peak–to–peak amplitude up to 7% of ozone variation from DIAL data at Hohenpeissenberg related to the 11–year solar cycle over the time period 1979 to early 2003.

Observational studies (Van Loon and Labitzke, 2000) indicate a large solar influence in the lower stratosphere as a dynamic result of the interaction in the upper stratosphere between ozone and UV radiation. With our data set we obtain a smaller
solar influence at the stratopause compared with the lower stratosphere. This makes sense since the increased absorption of solar irradiance during the solar maximum increases the temperature but also increases the production of ozone, which is the main absorber of radiation in this altitude region. In turn the temperature–dependent ozone photochemistry (anti-correlation between temperature and ozone) may partly compensate the solar irradiance influence at the stratopause. Kodera and Kuroda (2002) suggested that the impact the solar radiation has on the upper atmosphere and stratopause region influences the lower
stratosphere through modulation of the internal mode of variation in the polar night jet and a change in the Brewer-Dobson circulation. Various mechanisms have been proposed for the 11–year solar cycle influence on ozone variability. For instance, a change in mean meridional lower stratospheric dynamics between solar maximum and solar minimum may be the main factor for the ozone variability due to the solar activity signal. This change in ozone transport is supported by model and observational results (Mitchell et al., 2014; Calisesi and Matthes, 2007, and references therein). Model simulations (Gabriel et al., 2011) show
that zonal asymmetries in ozone, water vapour, and temperature fields are modulated by the 11–year solar cycle, providing a link between the solar cycle, zonal asymmetries, planetary waves, and the Brewer-Dobson circulation, all of them affecting the stratospheric ozone distribution (WMO, 2014).

The surprisingly strong ozone amplitude observed by GROMOS cannot be explained solely by the solar activity cycle. We hypothesise that the solar variability is in competition with other factors and together they emphasise the solar cycle response
in ozone at this station, e.g. strong ozone anomalies resulting from other factors may enhance the ozone anomaly related to





the solar activity cycle. Our study only considers a time interval of about 1.5 cycles of the solar activity. Thus, the derived 11–year amplitude could be influenced by ozone anomalies of other reasons which may accidentally occur during the solar maximum or the solar minimum phase. Gabriel et al. (2011) stated that the understanding of observed local changes may be improved if the change in regional wave patterns and associated mixing processes induced by the solar cycle are taken into

account. Moreover, there is a lack of long-term observational studies on the response of ozone to the solar cycle in the mid-latitudes. However the exact mechanism of the dynamical response to solar cycle variations is not fully understood and cannot be reproduced fully by chemistry-climate models (WMO, 2014). On the other hand, Offermann et al. (2015) have proposed self-sustained oscillators in the atmosphere for the first time to support that the oscillations they studied are excited internally in the atmosphere. They stated that the periods and phases observed might be interpreted as synchronisation effects that are

typical of non-linear oscillators. The connection with our study relies on the fact that both analysis are performed in Europe, in the middle atmosphere and at the same periods of oscillation. A self-excited oscillator might explain our strong amplitude in the response of ozone related with the 11–year period. Nevertheless, this is a novel interpretation which needs further analysis and with our experimental data we cannot prove the theory.

## 5.6  El Niño–Southern Oscillation (ENSO)

El Niño–Southern Oscillation (ENSO) is the globally dominating mode of inter-annual climate variability and an important driver for modulating the stratospheric climate in the Northern Hemisphere (NH). ENSO-forced variations in tropical upwelling lead to temperature and water vapour variations in the tropical lower stratosphere and impact the chemistry and transport of ozone. Atmospheric teleconnections lead to ENSO-related effects on the strength of planetary waves and the Brewer-Dobson circulation, both affecting the distribution of stratospheric ozone at middle and high latitudes (WMO, 2014). Particularly at

mid-latitudes, the ENSO effects have a longitudinal variability (Fischer et al., 2008). Observational studies show that warm ENSO events are associated with a weak and warm polar vortex over the Arctic during wintertime (van Loon and Labitzke, 1987). During the warm ENSO winters, enhanced planetary wave activity propagates from the troposphere to the stratosphere where it decelerates the zonal mean flow and accelerates the meridional (poleward) transport (Brönnimann, 2007).

We used the Multivariate ENSO index (MEI) as a proxy of the El Niño–Southern Oscillation (ENSO) signal. The MEI index

monitors the ENSO phenomenon with six variables (sea-level pressure, zonal and meridional components of the surface wind, sea surface temperature, surface air temperature, and total cloudiness fraction of the sky). Due to the delayed response of the ENSO signal in the northern extra tropics atmosphere we have performed the regression analysis with a time lag of 1–season in the MEI index. The northern ENSO effects manifest in late winter–early spring, i.e. 1–season delayed regarding the tropical ENSO, this is supported by observational and modelling studies (e.g., Ineson and Scaife, 2009; Manzini, 2009; Brönnimann

et al., 2007; Sassi et al., 2004; Brönnimann et al., 2004).

In Figure 8 is represented the amplitude of the ozone-ENSO for the time interval since January 1997 to January 2015. Even though the amplitude of the ozone-ENSO is quite small, for most of the altitude range around 1% (red line in Figure 1) we can clearly observe the response in ozone at northern mid-latitudes due to ENSO, especially the strong ENSO event in 1997-98 (Figure 8). The main feature is the opposite phase between the amplitudes of ozone-ENSO in the lower and middle stratosphere.



The green line in Figure 8 represents the MEI index, which is positive during the warm El Niño phase (wENSO) and negative during the cold La Niña phase (cENSO). The amplitude of ozone-ENSO decreases (increases) during wENSO (cENSO) in the lower stratosphere while it increases (decreases) in the middle stratosphere. This feature is also observed over Europe in the study of Fischer et al. (2008) by the SOCOL chemistry-climate model. The mechanism which explains this pattern is a strengthening of the residual mean circulation caused by anomalously vertically propagating waves (Fischer et al., 2008, and references therein).

## 6 Conclusions

Stratospheric ozone profiles for the period 1997–2015 from GROMOS were used to evaluate the effects on ozone due to naturally induced oscillations. To perform the study, we used a multilinear parametric regression model. We derived the behaviour of the natural oscillations in stratospheric ozone which are mainly induced by the annual and semi-annual cycle, the QBO, ENSO and the solar activity cycle. The effects on middle and upper stratospheric ozone are primarily caused by annual and semi-annual oscillations. Our observed AO dominates in summertime at 10 hPa (32 km) and in wintertime over 2 hPa (42 km) with amplitudes around 1 ppm and 0.6 ppm respectively. The SAO has its maximum amplitude of approximately 0.4 ppm in springtime at 3 hPa (40 km). We found a link between the upper stratospheric SAO in winter and the polar winter stratopause warmings since the temperature enhancement due to these warmings lead to upper stratospheric ozone depletion. At this altitude range, the ozone variability is mainly controlled by photochemistry since its lifetime is relatively short. Therefore, the main causes of natural oscillations in ozone in the middle and upper stratosphere arise from the temperature–dependent ozone photochemistry. On the other hand, the effects observed in the lower and middle stratosphere are owed to inter-annual variations of the atmospheric circulation and the solar radiation. As inter-annual variations we consider the QBO, ENSO and the solar activity cycle. In the lower and middle stratosphere, the long lifetime of ozone is responsible for changes in ozone that are predominantly due to transport. We identify the induced meridional circulation driven by the different phases of the aforementioned inter-annual variations as primary source of effects in ozone due to transport processes. The planetary wave propagation is affected by these inter-annual variations and in turn will influence the strength of the mean meridional circulation. The effect of QBO in lower stratospheric ozone increase (decrease) during the westerly (easterly) phase of the Singapore zonal winds. Thereby, the response of stratospheric ozone to the QBO above the mid-latitude station in Bern is opposed to that of tropical stratospheric ozone. The amplitude of the ozone-QBO at Bern is approximately 3% in the lower stratosphere. On the other hand, we note an increase (reduction) in ozone during the solar maxima (minima) period. The strong amplitude of about 5% for the observed solar cycle effect at Bern might be partly due to accidentally superposed ozone anomalies of meteorological reason or to an unknown amplification mechanism. The observed influence of ENSO on stratospheric ozone above Bern has an amplitude up to 1%. During wENSO (cENSO) our results show a diminution (rise) in ozone mixing ratios in lower stratosphere. On the contrary, in the middle stratosphere we notice an increase (decrease) for wENSO (cENSO). Our empirical results are in agreement with other model and observational studies. Nonetheless, with this study we present a new perspective



of the natural oscillation of stratospheric ozone from a northern mid-latitude station, which is very useful for the identification of regional effects and also for the validation of other studies.

*Acknowledgements.* This work was supported by the Swiss National Science Foundation under Grant 200020 - 160048 and MeteoSwiss GAW Project: "Fundamental GAW parameters measured by microwave radiometry".



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





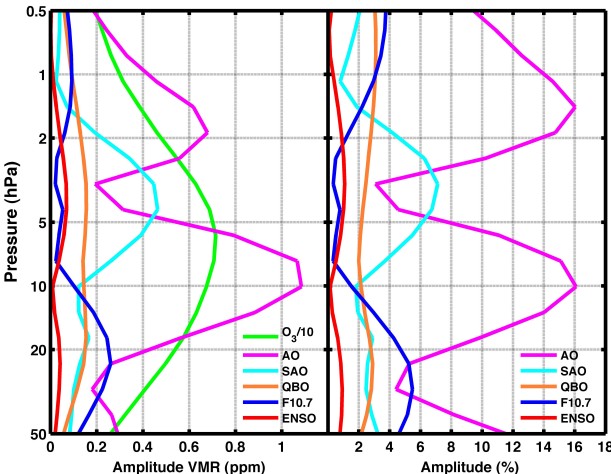

**Figure 1.** Amplitudes of the natural oscillations in stratospheric ozone derived by multilinear regression from the GROMOS observations at Bern (1997-2015). The ozone mean profile divided by 10 is shown by the green line.

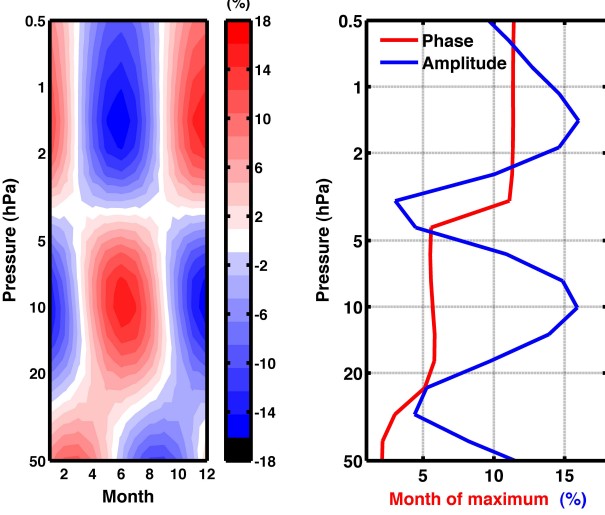

**Figure 2.** The amplitude of the ozone-AO depending on the months of the year is represented in % in the left panel. The profile of the amplitude of ozone-AO in % (blue line) and its phase given as the month of the maximum (red line) is shown in the right panel.





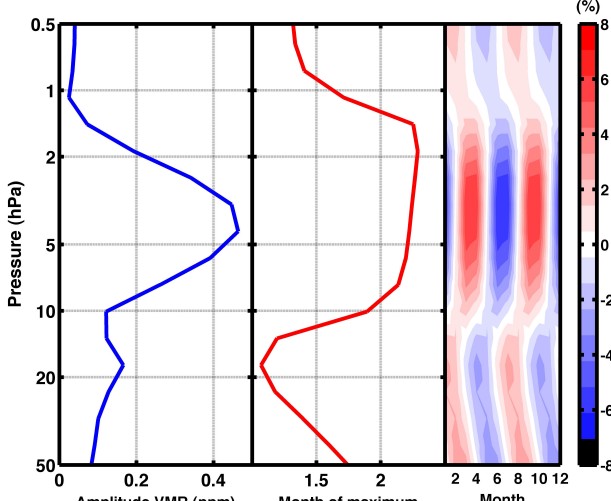

**Figure 3.** The profile of the amplitude of ozone-SAO in ppm (blue line) and its phase given as the month of the maximum (red line) are shown in the left and middle panel. The amplitude of the ozone–SAO depending on the months of the year is represented in % in the right panel.

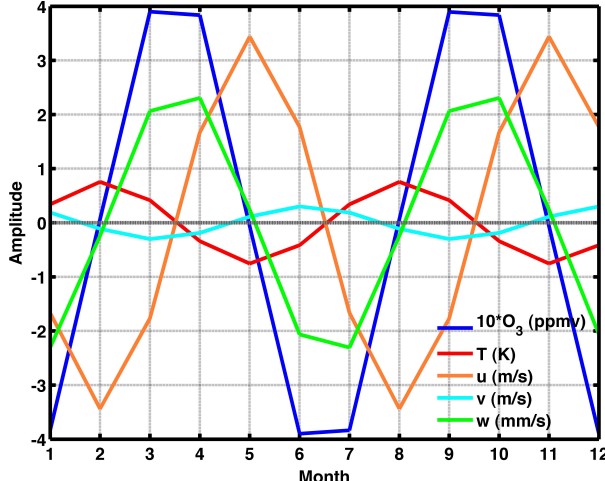

**Figure 4.** Amplitude of the temperature-, zonal wind-, meridional wind-, vertical wind-, ozone-SAO throughout the year at 3hPa. The ozone-SAO is multiplied by 10.



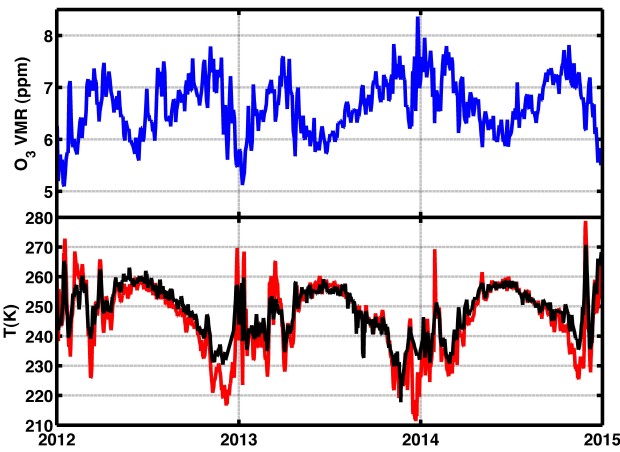

**Figure 5.** Ozone VMR from GROMOS (blue line), temperature from TEMPERA (black line) and ECMWF (red line) at 3 hPa for the period from January 2012 to January 2015

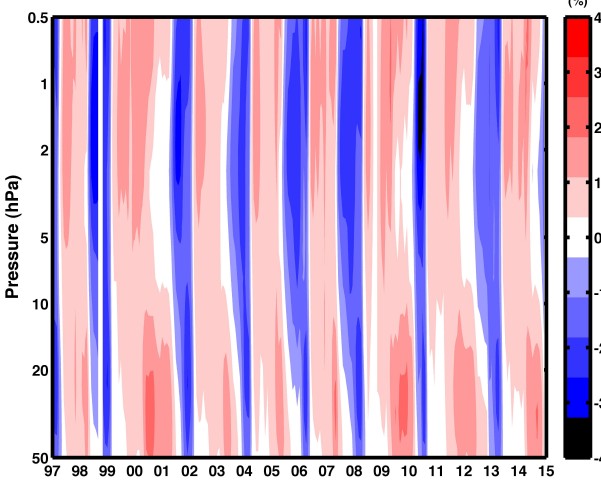

**Figure 6.** Amplitude of the ozone-QBO determined by multilinear regression for the time interval from January 1997 to January 2015.

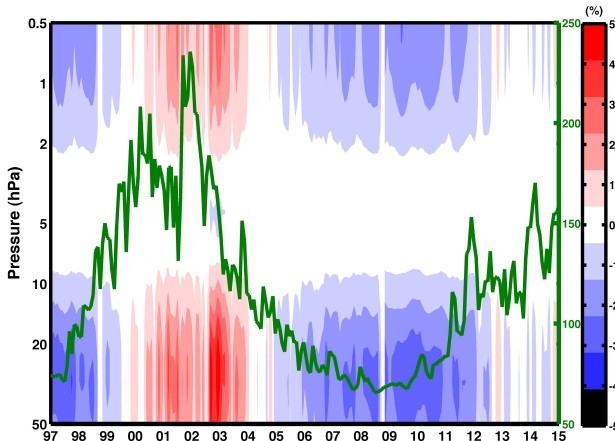

**Figure 7.** Amplitude of the ozone-solar F10.7 flux fitted by multilinear regression for the time interval from January 1997 to January 2015. The green line represents the time series of the monthly mean 10.7 cm solar radio flux as a measure of the solar activity.

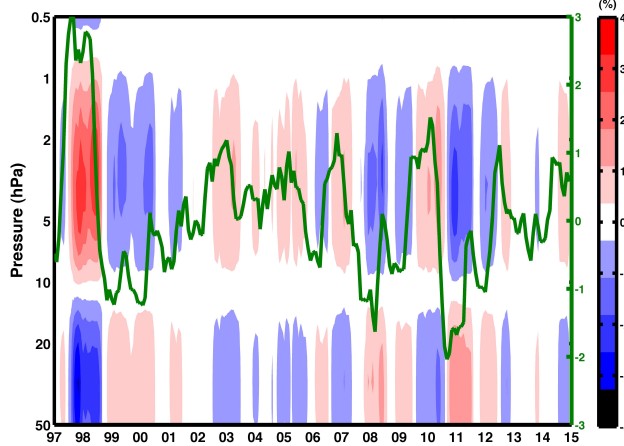

**Figure 8.** Amplitude of the ozone-ENSO fitted during multilinear regression for the time interval from January 1997 to January 2015. The green line represents the MEI index.