# Peer review of "The natural oscillations in stratospheric ozone observed by the GROMOS microwave radiometer at the NDACC station Bern"

_Atmospheric Chemistry and Physics, 2016_

## Referee Comment (RC1) · E. Remsberg (Referee) · 14 Apr 2016

General comments:

The authors analyze time series of ground-based, millimeter-wave ozone spectrometer profiles from 50 to 0.5 hPa at Bern, Switzerland, and relate the components of their variability to known atmospheric forcing mechanisms. Their analyzed semi-annual, annual, and QBO-like variations are in good agreement with those in the literature, as obtained from other data sets and with models. While those findings are not new, I suggest keeping them in the manuscript for completeness. The observed ozone responses to ENSO and solar cycle forcings ought to be of more interest to readers. The authors offer reasonable explanations for those longer-term responses. Overall, the manuscript is organized and written well. Figures are appropriate and illustrate the results clearly. The present manuscript is complementary to analyses of ozone trends by Moreira et al. (2015), who made use of the same data time series.

Specific comments:

p. 2, line 12—It would be helpful for the authors to mention one or more of the "open questions" (or give a reference to them) that they hope to address with their study.

p. 2, line 23—Mitchell et al. is concerned mainly with re-analyses of temperature, not ozone.

p. 2, line 26—As I am sure that the authors are aware, it is easier to be certain about a specific forcing mechanism from analyses of ozone data across multiple latitudes/longitudes, e.g., as from the analyses of satellite data by Nair et al. and Yang et al.

p. 3, line 13—The rather low vertical resolution of the profile data makes it difficult to sort out the effects of transport from meridional mixing versus that due to the Brewer-Dobson circulation or to resolve properly the ozone response to a solar cycle forcing in the upper stratosphere.

p. 8, line 24—"A possible role…" is not a complete sentence.

p. 10, lines 1 to 7—There is not a 1-year lag in the response of upper stratospheric ozone to the solar uv-flux forcing (Cunnold et al., 2004). Analyses for a solar cycle response in ozone are best done using the observed solar flux as a proxy, and Steinbrecht et al. seem to agree in their Reply (JGR, 2004, D14036, although not cited here). The analyses in the present manuscript are appropriate because they make use of time series of the F10.7 flux.

p. 10, lines 18ff—With this paragraph it should become clear to the reader that it is next to impossible to know about the cause(s) of decadal-scale forcings in time series of middle to lower stratospheric ozone from a single middle latitude station. To their credit, the authors seem to be acknowledging that difficulty in their discussion.

p. 10, line 33—"surprisingly strong ozone amplitude"…Are the authors referring to the 5% amplitude of lower stratospheric ozone in Figure 1?

---

## Referee Comment (RC2) · Anonymous Referee #2 · 20 Apr 2016

The paper of Moreira et al. investigates natural oscillations in stratospheric ozone observed by the GROMOS microwave radiometer at the NDACC station Bern (January 1997 to January 2015). The variability of these data is fitted using a regression analysis covering the annual oscillation, the semi-annual oscillation as well as QBO, ENSO, and solar cycle. The paper presents some new results, particularly with respect to ENSO and solar cycle effects. It should therefore be interesting for the ACP readership. However, before publication some important revisions should be implemented as detailed below. In particular, it would be useful to include a few direct comparisons of observations with analyses results (in addition to Figure 5). My main concerns are related to the robustness of the results, particularly of the results obtained for the solar cycle.

Major 1: The authors should provide more information on the robustness of their results. In this context, they should show a few comparisons between the time serious of the original data and the regression results (sum and individual components), for example, at the altitudes of maximum AO and SAO amplitudes and at altitudes of relatively large solar and ENSO signals. This would give an impression, how well the observed variability is fitted and what is left as residuals.

Major 2: The authors state that the surprisingly large amplitudes of the solar cycle signal cannot be solely explained by solar cycle effects alone but may be increased by a self excited oscillation. This is a rather speculative argument. In addition, it is not clear why the effect of such a self excited oscillation can be fitted with a same time series of normalized solar radio flux at 10.7 cm (shifted by 1 year). This procedure also inherently assumes that the effect of the self excited oscillation would how the same time behavior (phase shift) as the other solar effects. Is this really plausible? Again, it would be good to analyze the robustness of the results obtained from scaling one particular proxy curve. What is the impact of data anomalies? What is the impact of vertical resolution?

Minor: What is the altitude grid used in the fitting procedure?

Page2, line5, Please add a reference after "... eddy mixing processes".

Page2, line13: typo? Please replace "roll" by "role".

Page3, line14 (and in the following), "measurement contribution" is more common than "measurement response" in this context.

Page3, line13: I would not state that the ground station data have been validated by satellite observations. Usually, the opposite is the case. Maybe "... are in good agreement" with satellite observations.

Page3, line23: Maybe specify "... to analyses the temperature ... for natural variability are from the European Centre for Medium-Range Weather Forecast /ECMWF)."

Page3, line39: "molecular oxygen"?

Page4, line4: The choice of the indices for the last term is somewhat confusing. It is not immediately obvious that AO and SAO are represented.

Page4, line8: The expression "1-year" for a 1-year time shift and "1-season" for a shift by 1 season is not immediately obvious.

Page5, line16: Please reformulate first sentence.

Page 5, line19: "Local time" is more common than "time of day".

Page6, line2: Could you please explain, why this is a "consequence"?

Page 6, line11: What do you mean? The 11-year oscillation is the result of several processes.

Page 6, line 15: Why "Otherwise"?

Page 6, line22-23: For the same latitude?

Figure 4 and discussion in text.: The y-axis of Figure 4 does show the amplitude but the cycle. In addition, the x-axis should have a length corresponding to 12 months instead of 11 months.

Page 7: line30: The ozone SAO and the temperature SAO seem to be 1.5 months out of phase (and not 1). This becomes more apparent, when not looking only at the under-sampled maxima.

Page 8, line1: Maybe replace "equinox season" by "around equinox.

Page 8, line6-7: Is this only an effect of chemistry? Transport should also play a role (considering the vertical ozone gradient).

Page 9, line11: Maybe replace "relatively in agreement" by "in relatively good agreement".

---

## Author Comment (AC1) · 23 May 2016

**Response to the referee Ellis Remsberg**

Lorena Moreira

May 23, 2016

We are very grateful to Ellis Remsberg (Referee #1) for the careful reading of our manuscript and for providing very constructive comments which certainly helped to improve the manuscript. This document includes all the referee's comments as well as our replies to every one of them.

**Specific comments**

1. **Comments from the referee:** p. 2, line 12 – It would be helpful for the authors to mention one or more of the "open questions" (or give a reference to them) that they hope to address with their study.

   **Author's response:**
   Two examples of the open questions mentioned in the manuscript are for instance the effect on ozone due to ENSO. Fischer *et al.* (2008) found a pattern of reduced ozone over the tropics together with increased ozone and weakened zonal flow at higher latitudes in the stratosphere. This pattern suggests a strengthening of the residual mean circulation caused by anomalously vertical propagating waves. But this pattern is only observed in the SOCOL model, i.e. the cause of the effect can only be explained by one particular model therefore is still an open question. Another example of an open question is the amplitude of the ozone response to the solar activity cycle. For example, Ball *et al.* (2016) have found that model simulations of the ozone response to solar variability are inconsistent with satellite observations. Our study with data of a single station located in central Europe is useful to validate the variability of stratospheric ozone related to some natural oscillations and thus be able to confirm the proposed mechanisms responsible for the observed effects.

   **Author's changes in the manuscript:** p. 2, line 13
   Even though a number of mechanisms have been proposed as interpretations of the natural ozone variations in previously mentioned analysis, there are still open questions in the attribution of the causes to the effects observed in the stratosphere. For instance, discrepancies between different model simulations (e.g. Fischer *et al.*, 2008) or inconsistencies between model simulations and observations (e.g. Ball *et al.*, 2016) in the response of ozone to a natural oscillation complicates the understanding of the cause of the observed effect.

2. **Comments from the referee:** p. 2, line 23 – Mitchell et al. is concerned mainly with re-analyses of temperature, not ozone.

**Author's response:**
We agree with the referee that Mitchell *et al.* (2014) is concerned with temperature, but it was used as an example about how different data sets can give different results. We have changed the reference from Mitchell *et al.* (2014) by the reference from Ball *et al.* (2016), since it is more appropriate for our study.

**Author's changes in the manuscript:** p. 2, line 26
An analysis of this sort at a single station may offer valuable information, useful not only for the comprehension at regional levels but also for the validation of model simulations. In fact, Ball *et al.* (2016) have found that model simulations of the ozone response to solar variability are inconsistent with satellite observations.

3. **Comments from the referee:** p. 2, line 26 – As I am sure that the authors are aware, it is easier to be certain about a specific forcing mechanism from analyses of ozone data across multiple latitudes/longitudes, e.g., as from the analyses of satellite data by Nair et al. and Yang et al.

**Author's response:**
We support the referee's assertion that it is easier to be certain about a specific forcing mechanism from analyses of ozone data across multiple latitude/longitudes, although data from ground-based stations are crucial to be certain about regional effects, and ground-based microwave radiometry has a dense sampling in local solar time.

**Author's changes in the manuscript:** p. 2, line 29
In addition, our station can contribute to the understanding of the natural oscillations since there are just a few observational studies based on ground-stations (e.g., Nair *et al.*, 2013; Calisesi *et al.*, 2005; Schneider *et al.*, 2005) of naturally induced stratospheric ozone variability at mid-latitudes.

4. **Comments from the referee:** p. 3, line 13 – The rather low vertical resolution of the profile data makes it difficult to sort out the effects of transport from meridional mixing versus that due to the Brewer-Dobson circulation or to resolve properly the ozone response to a solar cycle forcing in the upper stratosphere.

**Author's response:**
The referee is right to point out this, yet it is still possible to observe different effects in the lower, middle and upper stratosphere and the lower mesosphere due to different forcings mechanism and quantify them, and therefore validate the amplitudes obtained by model simulations and/or measured by satellites observations.

**Author's changes in the manuscript:** No changes.

5. **Comments from the referee:** p. 8, line 24 – "A possible role..." is not a complete sentence.

**Author's response:**
Thanks for spotting. We have corrected this.

**Author's changes in the manuscript:** p. 9, line 2
A possible role of stratospheric warmings for the generation of the ozone-SAO at mid-latitudes was previously mentioned by Perliski *et al.* (1989) and Maeda (1984).

6. **Comments from the referee:** p. 10, lines 1 to 7 – There is not a 1-year lag in the response of upper stratospheric ozone to the solar uv-flux forcing (Cunnold et al., 2004). Analyses for a solar cycle response in ozone are best done using the observed solar flux as a proxy, and Steinbrecht et al. seem to agree in their Reply (JGR, 2004, D14036, although not cited here). The analyses in the present manuscript are appropriate because they make use of time series of the F10.7 flux.

   **Author's response:**
   We agree on the referee's comment. The text has been modified according to it.

   **Author's changes in the manuscript:** p. 10, line 12
   Stratospheric ozone has a delayed response to the solar activity cycle (e.g., Steinbrecht *et al.*, 2004;  ; Lee and Smith, 2003).

7. **Comments from the referee:** p. 10, lines 18ff – With this paragraph it should become clear to the reader that it is next to impossible to know about the cause(s) of decadal-scale forcings in time series of middle to lower stratospheric ozone from a single middle latitude station. To their credit, the authors seem to be acknowledging that difficulty in their discussion.

   **Author's response:**
   We acknowledge that with time series of middle to lower stratospheric ozone from a single middle latitude station we are not able to know about the cause(s) of the decadal-scale forcings. We by no means intend to diagnose global processes using data from a single location. But, in turn, the global processes should contribute to the explanation of the local behaviour. We can observe the ozone response and quantify it and therefore give evidence of this decadal-scale forcing over central Europe. Indeed, Ball *et al.* (2016) have shown inconsistencies between model simulations and satellites observations in the response of ozone to solar activity cycle. Further, we found very good agreement with the ozone amplitude observed by Steinbrecht *et al.* (2004) with a ground-based station located in Germany.

   **Author's changes in the manuscript:** No changes.

8. **Comments from the referee:** p. 10, line 33 – "surprisingly strong ozone amplitude"... Are the authors referring to the 5% amplitude of lower stratospheric ozone in Figure 1?

   **Author's response:**
   Yes, we are referring to the 5% amplitude of lower stratospheric ozone due to the solar activity cycle.

   **Author's changes in the manuscript:** p. 11, line 9
   The surprisingly strong ozone amplitude observed by GROMOS (5% in the lower stratosphere) cannot be explained solely by the solar activity cycle.

**Response to anonymous referee #2**

Lorena Moreira

May 23, 2016

We are very thankful to the anonymous Referee #2 for the evaluation of our manuscript and for the valuable comments that helped significantly to improve the manuscript. We have revised the manuscript by following each one of your suggestions. Below we try to answer each comment.

**Major comments**

1. **Comments from the referee:** The authors should provide more information on the robustness of their results. In this context, they should show a few comparisons between the time series of the original data and the regression results (sum and individual components), for example, at the altitudes of maximum AO and SAO amplitudes and at altitudes of relatively large solar and ENSO signals. This would give an impression, how well the observed variability is fitted and what is left as residuals.

   **Author's response:**
   The annual oscillation appears in the middle and upper stratosphere whereas the semi-annual oscillation maximum is found in the upper stratosphere. On the other hand the QBO and ENSO are present in the lower and middle stratosphere. The solar cycle effect is mainly observed in the lower stratosphere. Therefore in order to show a comparison between the time series of the original data and the regression results we have selected 3 pressure levels 23, 10 and 3 hPa which are representatives of the lower, middle and upper stratosphere. The ozone monthly means measured by GROMOS and the calculated fit are the blue and red lines respectively in the first panel of each column. In the second, third and fourth panels of every column are shown the ozone fitted signals of the proxies QBO (magenta line), solar F10.7 cm flux (red line) and ENSO (green line). Finally in the lowermost panels are represented the residuals. The residuals are within 0.5 ppm except for some special cases. In the lower stratosphere the regression model explains about 50% of the variance whereas in the middle an upper stratosphere it explains around 80%.

   **Author's changes in the manuscript:** Page 4, line 25
   The Figure 1 shows 3 examples of the fit in the lower, middle and upper stratosphere (23, 10 and 3 hPa). The ozone monthly means measured by GROMOS (blue line) and the calculated fit (red line) are represented in the first panel of each column. In the second, third and fourth panels of every column are shown the ozone fitted signals of the proxies QBO (magenta line), solar F10.7 cm flux (red line) and ENSO (green line). Finally in the lowermost panels are represented

[Figure]

Figure 1: In the first panel of each column is represented the GROMOS monthly means in blue and the calculated fit in red. Every column represents a different pressure levels, representative of the lower, middle and upper stratosphere (23, 10 and 3 hPa, respectively). In the second, third and fourth panel of each column are shown the ozone fitted signals of the proxies QBO (magenta line), solar F10.7 cm flux (red line) and ENSO (green line). The last panel of every column shows the residuals.

the residuals. The residuals are within 0.5 ppm except for some special cases. In the lower stratosphere the regression model explains about 50% of the variance whereas in the middle an upper stratosphere it explains around 80%.

2. **Comments from the referee:** The authors state that the surprisingly large amplitudes of the solar cycle signal cannot be solely explained by solar cycle effects alone but may be increased by a self excited oscillation. This is a rather speculative argument. In addition, it is not clear why the effect of such a self excited oscillation can be fitted with a same time series of normalised solar radio flux at 10.7 cm (shifted by 1 year). This procedure also inherently assumes that the effect of the self excited oscillation would show the same time behaviour (phase shift) as the other solar effects. Is this really plausible? Again, it would be good to analyse the robustness of the results obtained from scaling one particular proxy curve. What is the impact of data anomalies? What is the impact of vertical resolution?

**Author's response:**

We agree with the referee that it is a rather speculative argument, but the idea of this self-sustained mechanism is only meant as one more example for the unknown explanation of the strong ozone amplitude due to the solar activity cycle.

The solar cycle effect in ozone has its maximum amplitude around 20 hPa. The fitted signal of the proxy solar radio flux at 10.7 cm at 20 hPa is shown in the third panel of the left column in Figure 1. The residuals are smaller than 0.4 ppm during the solar maximum and solar minimum.

The impact of the vertical resolution is minor since the vertical resolution is fine

enough to distinguish between the lower and middle stratosphere.

**Author's changes in the manuscript:** No changes.

**Minor comments**

1. **Comments from the referee:** What is the altitude grid used in the fitting procedure?

   **Author's response:**
   Every 2 km the GROMOS pressure grid has a point.

   **Author's changes in the manuscript:** No changes.

2. **Comments from the referee:** Page 2, line 5, Please add a reference after "... eddy mixing processes".

   **Author's response:**
   No comments.

   **Author's changes in the manuscript:** Page 2, line 5
   Planetary waves play an essential role in driving the zonal mean transport by the Brewer-Dobson circulation (BDC) and eddy mixing processes (Gabriel *et al.*, 2011).

3. **Comments from the referee:** Page 2, line 13: typo? Please replace "roll" by "role".

   **Author's response:**
   No comments.

   **Author's changes in the manuscript:** Page 2, line 16
   Moreover, these modes of variability do not always play a role in isolation ...

4. **Comments from the referee:** Page 3, line 14 (and in the following), "measurement contribution" is more common than "measurement response" in this context.

   **Author's response:**
   No comments.

   **Author's changes in the manuscript:** Page 3, line 17
   The measurement contribution between ...

5. **Comments from the referee:** Page 3, line 13: I would not state that the ground station data have been validated by satellite observations. Usually, the opposite is the case. Maybe "... are in good agreement" with satellite observations.

   **Author's response:**
   No comments.

   **Author's changes in the manuscript:** Page 3, line 21
   The vertical ozone profiles from GROMOS have been validated by means of nearby ozone sondes, ground-based stations and are in good agreement with satellite observations.

6. **Comments from the referee:** Page 3, line 23: Maybe specify "... to analyses the temperature... for natural variability are from the European Centre for Medium-Range Weather Forecast (ECMWF)."

   **Author's response:**
   No comments.

   **Author's changes in the manuscript:** Page 3, line 26
   The data used to analyse the temperature, zonal wind, meridional wind and vertical wind for natural variability are from the European Centre for Medium-Range Weather Forecast (ECMWF)

7. **Comments from the referee:** Page 3, line 39: "molecular oxygen"?

   **Author's response:**
   No comments.

   **Author's changes in the manuscript:** Page 3, line 32
   TEMPERA is a novel ground-based microwave radiometer that measures the thermal radiation emitted by molecular oxygen in the ...

8. **Comments from the referee:** Page 4, line 4: The choice of the indices for the last term is somewhat confusing. It is not immediately obvious that AO and SAO are represented.

   **Author's response:**
   The sum term comprises 2 sine and cosine functions with period length $l_n$, which represent the annual ($l_n$=12 months) and semi-annual cycle ($l_n$=6 months).

   **Author's changes in the manuscript:** Page 4, line 8

   $$\hat{y}(t) = a+b{\cdot}t+c_1{\cdot}qbo_1(t)+d_1{\cdot}qbo_2(t)+e{\cdot}F10.7(t)+f{\cdot}MEI(t)+\sum_{n=1}^{2}(g_n{\cdot}\sin(\frac{2\pi \cdot t}{l_n})+h_n{\cdot}\cos(\frac{2\pi \cdot t}{l_n}))$$

   Page 4 line 22
   The coefficients $a$, $b$, $c_1$, $d_1$ , $e$, $f$, $g_1$, $g_2$, $h_1$ and $h_2$ are fitted to the monthly means using the method of von Clarmann *et al.* (2010).

9. **Comments from the referee:** Page 4, line 8: The expression "1–year" for a 1–year time shift and "1–season" for a shift by 1 season is not immediately obvious.

   **Author's response:**
   No comments.

   **Author's changes in the manuscript:** Page 4, line 21
   In order to avoid this problem the fitting of these terms was made with a time shift (1–year delay for the solar activity cycle and 1–season delay for the ENSO)

10. **Comments from the referee:** Page 5, line 16: Please reformulate first sentence.

    **Author's response:**
    The term partitioning does not need to be defined, it is a quite well-established technical term.

**Author's changes in the manuscript:** Page 5, line 25
 The partitioning of odd oxygen depends upon the photolysis rate of ozone, the $O + O_2$ reaction rate coefficient and the air density.

11. **Comments from the referee:** Page 5, line 19: "Local time" is more common than "time of day".

    **Author's response:**
    No comments.

    **Author's changes in the manuscript:** Page 5, line 27
    The number of photons in turn depends upon a number of other parameters: altitude, latitude, season and local time.

12. **Comments from the referee:** Page 6, line 2: Could you please explain, why this is a "consequence"?

    **Author's response:**
    No comments.

    **Author's changes in the manuscript:** Page 6 line 10
    As a result of using them ...

13. **Comments from the referee:** Page 6, line 11: What do you mean? The 11-year oscillation is the result of several processes.

    **Author's response:**
    We mean that the amplitude of the 11–year oscillation (solar cycle effect) in stratospheric ozone observed by the GROMOS microwave radiometer could be enhanced by ozone anomalies of other reasons which may accidentally occur during the solar maximum or the solar minimum.

    **Author's changes in the manuscript:** Page 6, line 18
    However, the 11–year oscillation in stratospheric ozone observed by the GROMOS microwave radiometer could be influenced by interfering processes which we will discuss later.

14. **Comments from the referee:** Page 6, line 15: Why "Otherwise"?

    **Author's response:**
    No comments.

    **Author's changes in the manuscript:** Page 6, line 24
    On the other hand in the upper stratosphere ...

15. **Comments from the referee:** Page 6, line 22–23: For the same latitude?

    **Author's response:**
    Yes, for the same latitude.

    **Author's changes in the manuscript:** No changes.

16. **Comments from the referee:** Figure 4 and discussion in text.: The y-axis of Figure 4 does show the amplitude but the cycle. In addition, the x-axis should have a length corresponding to 12 months instead of 11 months.

**Author's response:**
    No comments.

**Author's changes in the manuscript:** Figure 4

[Figure]

Figure 5: Amplitude of the temperature-, zonal wind-, meridional wind-, vertical wind-, ozone-SAO throughout the year at 3hPa. The ozone-SAO is multiplied by 10.

17. **Comments from the referee:** Page 7, line 30: The ozone SAO and the temperature SAO seem to be 1.5 months out of phase (and not 1). This becomes more apparent, when not looking only at the under-sampled maxima.

    **Author's response:**
        No comments.

    **Author's changes in the manuscript:** Page 8, line 9
        The ozone-SAO is 1.5 months out of phase with that of temperature ...

18. **Comments from the referee:** Page 8, line 1: Maybe replace "equinox season" by "around equinox.

    **Author's response:**
        No comments.

    **Author's changes in the manuscript:** Page 8, line 14
        The cold phase descends with the westerly shear zone around the equinox.

19. **Comments from the referee:** Page 8, line 6–7: Is this only an effect of chemistry? Transport should also play a role (considering the vertical ozone gradient).

    **Author's response:**
        As written in the manuscript the out of phase relationship between temperature and vertical wind is expected due to adiabatic cooling associated with ascent of ozone-rich air.

    **Author's changes in the manuscript:** No changes.

20. **Comments from the referee:** Page 9, line 11: Maybe replace "relatively in agreement" by "in relatively good agreement".

   **Author's response:**
   No comments.

   **Author's changes in the manuscript:** Page 9, line 21

[revised manuscript text omitted]